# A Comprehensive Review of the Genetic and Epigenetic Contributions to the Development of Fibromyalgia

**DOI:** 10.3390/biomedicines11041119

**Published:** 2023-04-07

**Authors:** Erik A. Ovrom, Karson A. Mostert, Shivani Khakhkhar, Daniel P. McKee, Padao Yang, Yeng F. Her

**Affiliations:** 1Mayo Clinic Alix School of Medicine, Rochester, MN 55905, USA; ovrom.erik@mayo.edu; 2Department of Physical Medicine and Rehabilitation, Mayo Clinic Hospital, Rochester, MN 55905, USA; 3Department of Orthopedics and Rehabilitation, University of Wisconsin School of Medicine and Public Health, Madison, WI 53705, USA; 4Department of Psychiatry and Psychology, Mayo Clinic Hospital, Rochester, MN 55905, USA; 5Department of Anesthesiology and Perioperative Medicine, Mayo Clinic Hospital, Rochester, MN 55905, USA

**Keywords:** fibromyalgia, genetics, epigenetics

## Abstract

This narrative review summarizes the current knowledge of the genetic and epigenetic contributions to the development of fibromyalgia (FM). Although there is no single gene that results in the development of FM, this study reveals that certain polymorphisms in genes involved in the catecholaminergic pathway, the serotonergic pathway, pain processing, oxidative stress, and inflammation may influence susceptibility to FM and the severity of its symptoms. Furthermore, epigenetic changes at the DNA level may lead to the development of FM. Likewise, microRNAs may impact the expression of certain proteins that lead to the worsening of FM-associated symptoms.

## 1. Introduction

Fibromyalgia (FM) is a centrally and peripherally mediated chronic pain syndrome with biological, psychological, and environmental predispositions [1,2,3,4]. It is estimated that the prevalence of FM in the general population is 2% [5]. FM is characterized by generalized chronic pain, fatigue, sleep changes, decreased cognitive function, and numerous tender points throughout the body [6]. Diagnosing and treating FM are challenging. FM has a high comorbidity rate with rheumatologic disorders such as psoriatic arthritis and ankylosing spondylitis [7]. Many FM individuals have psychiatric disorders [8].

FM has a strong genetic component, and the risk of developing FM is eightfold higher among first-degree relatives, as evidenced by familial aggregation studies [9]. The discovery of decreased serum and cerebrospinal fluid levels of serotonin (5-HT) in FM patients has guided many genetic studies [10,11]. Similarly, the catabolism and anabolism of other neurotransmitters, such as dopamine, were examined [12,13]. This has expanded to genes involved in pain processing and inflammation that may amplify pain signals in the nervous system or systemically.

Multiple epigenetic mechanisms of gene regulation have been studied in the pathogenesis and symptomatology of FM, including micro-RNA and DNA methylation. Environmental factors in the development of FM are significant. At present, childhood trauma, physical abuse, and chronic psychosocial stressors are believed to amplify stress responses mediated by the hypothalamic pituitary axis, ultimately leading to higher concentrations of substance-P in the central nervous system and increased pain interference in day-to-day life [14,15,16]. In this review, we will not examine the role of the environment in the development of FM.

In the last ten years, there have been unprecedented advances in technology to identify genetic and epigenetic contributions to the development of FM. This has led to a better understanding of disease pathogenesis. In this review, we focus on the associations of genetic and epigenetic changes with the development of FM and its related symptoms.

## 2. Methods

### 2.1. Literature Search Strategy

This study protocol followed the methodology described in the PRISMA statement for systematic reviews [17]. A comprehensive search of several databases was performed on 4 September 2022. Results were limited to English Language. No date limits for the search were applied. Databases searched (and their content coverage dates) were Ovid MEDLINE(R) (1946+ including epub ahead of print, in-process, and other non-indexed citations), Ovid Embase (1974+), Ovid Cochrane Central Register of Controlled Trials (1991+), Ovid Cochrane Database of Systematic Reviews (2005+), and Scopus via Elsevier (1970+). The search strategies were designed and conducted by a medical librarian (L.C.H.) with input from the study investigators. Controlled vocabulary, supplemented with keywords, was used to search for studies of interest. The actual search strategy, listing all search terms used and how they were combined, is available in Appendix A.

### 2.2. Study Selection

A total of 1722 articles were screened by title, abstract, and full text by the authors (Y.F.H. and E.A.O.) (Figure 1). All studies concerning genetics and epigenetics in English for which full text was available were included in the review. Exclusion criteria comprised the following: review articles and conference proceedings. The first reviewer (Y.F.H.) extracted the following data from all included studies: (1) study, (2) gene, (3) polymorphism, (4) DNA methylation, (5) miRNA, (6) upregulation/overexpression, and (6) downregulation. This was later reviewed by the second reviewers (E.A.O., S.K. and K.A.M.).

## 3. FM Pathophysiology

The exact pathophysiology of FM is unknown. It is hypothesized that FM is a complex interaction between the peripheral nervous system (PNS), central nervous system (CNS), immune system, and psychosocial stressors [3]. In general, noxious input is detected in the peripheral nervous system by nociceptors and transmitted to the spinal cord and brain for interpretation. Negative feedback via the descending pathways can reduce the noxious signal. The pathways and mechanisms that modulate neuronal signal is referred to as central sensitization. Dysfunction in the pain pathways in the PNS, CNS, and immune system can influence central sensitization. Similarly, psychosocial stressors can accentuate pain and pain reactions.

In FM, there is increasing evidence of dysfunction in neurotransmitter metabolism leading to elevated levels of excitatory neurotransmitters in the ascending pain pathways and reduced levels of inhibitory neurotransmitters in the descending pain pathways, resulting in increased central sensitization [18]. Likewise, in the PNS, polymorphisms in voltage gated ions channels genes lead to altered noxious stimulus processing resulting in different pain phenotypes [19,20,21].

Inflammatory processes in the peripheral tissue, CNS, and brain have been implicated in the pathophysiology of FM [22]. It is known that noxious transmission is amplified in an inflamed state throughout the pain pathways leading to central sensitization. Studies have shown that there is an association between FM development and increased systemic inflammatory cytokines [23].

## 4. Genetic Contributions to the Development of FM

### 4.1. Catechol-O-Methyltransferase (COMT) Polymorphisms and FM

COMT is one of the primary enzymes that inactivates catecholamines, including dopamine, by transferring a methyl group from S-adenosyl-L-methionine to dopamine to generate 3-methoxytyramine. Functional polymorphisms in the *COMT* gene can change the enzyme’s activity to be either a fast or slow metabolizer of catecholamines [24]. The three common *COMT* genotypes are Val-158-Val, Met-158-Met, and Val-158-Met. The Val-158-Val genotype has the highest enzymatic activity. The Met-158-Met genotype has the lowest enzymatic activity [25]. Phenotypically, Met-158-Met carriers showed a higher threshold to pain and pressure stimuli [12,26,27] and fatigue levels than Met-158-Val carriers [28]. Val-158-Val carriers exhibited significantly worse working memory measures than Met-158-Val carriers [29]. Val-158-Val carriers also showed more pain catastrophizing thought and higher levels of anxiety and symptoms of depression than Met-158-Val carriers [30]. Val-158-Val carriers reported higher sensory and affective ratings of pain along with negative internal affective states [31].

In FM individuals, the *COMT* Val-158-Met genotype, also named rs4680, has been examined for its association with pain symptoms and mood disorders (Table 1). Multiple studies demonstrate that rs4680 increases the risk of developing FM [12,32,33,34,35,36]. However, repeat investigations show no association between rs4680 and an increased risk of developing FM [37,38,39,40]. A possible explanation is that rs4680 may be overrepresented within the FM population. Since rs4680 is associated with increased pain symptoms [12,32,33,36,41], it is not surprising that individuals with rs4680 report an increased severity of FM-associated symptoms.

Several other single nucleotide polymorphisms (SNP) were investigated for FM association in FM individuals from different countries. SNP rs4818 of the *COMT* gene was associated with FM risk in Korean [38], Brazilian [32], and Spanish populations [36]. However, rs4818 was not associated with FM diagnosis in a large study examining diverse FM individuals [42], the Mexican population [36], or the Spanish population [12]. Similar to rs4680, rs4818 is associated with pain sensitivity [32,38]. The presence of both rs4818 and a SNP (rs1799971) in the opioid receptor mu 1 (*OPRM1*) gene is associated with pain catastrophizing [41]. Additional variants have been evaluated, with some showing an association with FM development. SNP rs2097903 of the *COMT* gene is associated with a higher risk of FM susceptibility [43]. rs6269 is associated with FM development in a Spanish population [36]. rs4633 is associated with FM development in a Korean population [38].

### 4.2. Polymorphisms in 5-HT Processing and FM

The role of 5-HT in mood is well established. Low levels of 5-HT are associated with a low mood. Raising the level of 5-HT from low levels is associated with improved mood [44]. 5-HT also plays a role in pain modulation. Although the full relationship has not been fully elucidated, it appears that increased levels of 5-HT within the peripheral nervous system are associated with sensitization of peripheral nerves and hyperalgesia. In the central nervous system, the role of 5-HT in pain modulation is complicated. It is dependent on receptor availability and affinity, 5-HT concentration, spinal cord pathways, and associated cerebral neural networks [18]. Low 5-HT levels are associated with FM diagnosis [45,46].

To understand the associations between 5-HT and FM, a review of 5-HT biosynthesis and physiology is needed (Figure 2). Tryptophan hydroxylase converts tryptophan into 5-hydroxytryptaminae (5-HTP). Aromatic decarboxylase converts 5-HTP into 5-HT. 5-HT is transported to storage in the presynaptic vesicles by the vesicular monoamine transporter (SLC18A2: Solute carrier family 8A member 2). When 5-HT is released into the synaptic cleft, it interacts with post-synaptic 5-hydroxytryptamine receptors (5-HTR1, 2, 3A, 4, 6, and 7) to activate secondary messenger cascades. Simultaneously, 5-HT stimulates presynaptic 5-HTR1 in a negative feedback loop to inhibit further release of 5-HT and interacts with solute carrier family 6 member 4 (SCL6A4) to transport synaptic 5-HT back into the pre-synaptic neuron [47]. 5-HT is also transported back into the pre-synaptic neuron by the serotonin transporter (5-HTT) [48]. Inhibitory serotonergic 5-HT1A receptors in the presynaptic neurons become activated and decrease serotonergic signalling [49].

Any changes in the described pathway that affect 5-HT concentrations have been reported to increase FM susceptibility or alter FM-associated symptoms (Table 2). Individuals carrying a “short allele” polymorphism (44 base pair deletion) in the 5′ regulatory region of *SLC6A4*, also known as the serotonin transporter promoter region (*5-HTTLPR*), appear to have an increased risk of developing FM [50]. This polymorphism decreases SCL6A4 transporter expression. A potential mechanism is that an impaired ability for 5-HT reuptake by SLC6A4 increases 5-HT1 receptor-mediated negative feedback to decrease 5-HT concentration in the synaptic cleft [45,51]. Individuals carrying the short allele of the *5-HTTLPR* gene also have associated depression and anxiety disorders [50]. In a study examining the association between 5-HT autoantibodies and FM in a cohort of FM individuals, 73% of FM individuals have auto-antibodies to 5-HT [52]. They have lower 5-HT concentrations. It is postulated that a lower 5-HT concentration increases the risk of developing FM.

Polymorphisms in the *5-HTR2A* (5-hydroxytryptamine receptor 2A) gene have been associated with FM and depressive symptoms. FM female individuals carrying the *5-HTR2A* polymorphism of Cytosine-Thymine (CT) genotype have lower pain thresholds than those with TT and CC genotypes [41]. FM individuals with a combined *5-HT1a* CC and *5-HTT*-high expression have the fewest depressive symptoms compared to individuals with a combined *5-HT1a* CC/G and *5-HTT*-low expression [53].

SNP rs1062613 with the CC homozygotes genotype in the *5-HTR3a* gene were found to be more frequent in individuals with FM than in healthy controls. Carriers of the CC genotype were found to have fewer dopamine receptors available when compared to TT carriers, suggesting that CC carriers may experience less reward associated with dopamine release. However, there was no association between rs1062613 and pain threshold or tolerance [54,55]. Further examination of six different *5-HTR3a* variants and eight different *5-HTR3b* variants revealed mixed results when compared to healthy controls [55]. Several variants were more frequently observed in individuals with fibromyalgia compared to healthy controls but did not reach statistical significance.

### 4.3. Polymorphisms in Pain Processing and FM

The pain pathway consists of four general processes: transduction, transmission, modulation, and perception. Polymorphisms in genes involved in the processing of noxious stimuli along this pathway confer different pain phenotypes to individuals with FM (Table 3). The *ATP2C1* (ATPase Secretory Pathway Ca2+ Transporting 1) gene encodes a magnesium-dependent calcium pump protein called hSPCA1 (human secretory pathway Ca^2+^/Mn^2+^  ATPase protein 1), which mediates uptake of cytosolic calcium and magnesium to the Golgi apparatus [20]. A GWAS (genome-wide association study) discovered that a SNP (single nucleotide polymorphism) at the *ATP2C1* locus (rs10490825) is associated with chronic widespread pain [21]. This finding suggests that changes in intracellular calcium concentrations may play a role in pain transduction for FM individuals. Similarly, Andolina et al. reported that the *TRPM6* (transient receptor potential cation channel subfamily M member 6) SNP (rs395357), which encodes for a calcium and magnesium channel in pain transduction, is associated with a higher risk of developing FM [19,56]. HAP1 (Huntingtin-associated protein 1) is another intracellular protein involved in pain transduction. It is enriched in neuronal cells and mediates vesicular transport between organelles. A GWAS comparing FM and healthy volunteers showed that a *HAP1* SNP (rs4796604) is associated with a lower nociceptive flexion reflex threshold—the reflex by which individuals withdraw their hand from a hot oven dish before perceiving the painful stimulus at the supratentorial level [57]. The transient receptor potential vanilloid (TRPV) family of non-selective cation channels have a well-established role in sensing and transmitting noxious stimuli via sensory afferents [58]. Park et al. showed that a polymorphism of *TRPV3* (rs395357) is associated with the severity of fatigue symptoms and mental health in FM individuals [59].

In the pain transmission process, polymorphism in the *SNAP25* (Synatposome Associated Protein 25) gene is associated with worsening FM related symptoms in FM individuals. SNAP25 regulates neurotransmitter release via vesicle docking and fusion in the presynaptic neuron [65]. Balkarli et al. showed that FM individuals with the SNAP-25 TC genotype have higher Beck depression scale and visual analogue scale scores compared to FM patients with the TT and CC genotypes [94]. Similar to *SNAP25*, *SCN9A* (Sodium voltage gated channel alpha subunit 9) plays a role in mood symptoms and pain interference in FM individuals [95]. *SCN9A* encodes the Nav1.7 sodium channel, which is highly expressed in the dorsal root ganglion and sympathetic root ganglion. Polymorphisms in *SCN9A* (rs6754031 and rs4453709) are associated with higher FM impact questionnaire scores [63] and reduced motivation and activity [64].

Polymorphisms in proteins involving pain modulation and perception are associated with FM susceptibility and FM-related symptoms. Guanosine triphosphate cyclohydrolase 1 (GCH1) is needed for tetrahydrobiopterin (BH4) production. BH4 is a cofactor in dopamine and serotonin biosynthesis, which plays a role in the passage of pain signals between interneurons in the spinal cord and brain [14]. *GCH1* polymorphisms (rs3783641, rs84, rs752688, and rs4411417) have been reported to be associated with lower pain sensitivity and susceptibility to FM [66]. A polymorphism of the mu-opioid receptor gene (*OPRM1*; rs1799971) was shown to alter frontoparietal network processing during pressure stimulation in FM individuals [67]. Smith et al. reported that *TAAR1*, *RGS4*, and *GRIA4* (trace amine-associated receptor 1, regulator of G protein signaling 4, and glutamate ionotropic receptor AMPA type subunit 4, respectively) are associated with susceptibility to FM [96]. *TAAR1* encodes for a G-protein-coupled receptor that plays a role in reward and cognitive function [61]. RGS4 is a regulatory GTPase activating molecule that can negatively regulate G protein signaling in cells [97]. GRIA4 is a glutamate receptor that mediates excitatory neurotransmission [98]. Sleep disturbances are well established in FM individuals, and it has long been disputed whether these changes are in response to living with chronic pain, comorbid mental illness, or the underlying biological aberrations. Indeed, polymorphisms in several adrenergic receptor genes have been found in FM patients who suffer from disrupted sleep. The Gly16Arg SNP at the beta-2-adrenergic receptor gene has been found in patients with FM who suffer from sleep dysfunction [62,93]. Two SNPS in the alpha(1A)-adrenergic receptor gene (rs1048101 and rs1383914) were reported to be associated with the presence of FM and elevated FM impact questionnaires for disability [93]. The SNP rs574584 was associated with FM impact questionnaires measuring morning stiffness and tiredness upon awakening.

### 4.4. Inflammatory Genes/Proteins and FM

Inflammation is often associated with pain amplification. FM individuals appear to have higher levels of serum cytokines, chemokines, reactive oxygen species, and acute phase proteins [69], and there is increasing evidence that inflammation in FM may have a genetic underpinning (Table 3).

Interleukins are a class of cytokines that facilitate communication between white blood cells during an inflammatory response [70], and serum levels of many interleukins have been found to be elevated in FM individuals. An exome sequencing of 19 probands in a nuclear family of FM showed two nonsense mutations in chromosome 11 putative open reading frame 40 (*C11orf40)* and zinc finger protein 77 (*ZNF77*) with increased transmission to affected probands. These two nonsense mutations were associated with elevated plasma levels of inflammatory cytokines compared to controls [71]. Yigit et al. showed that an interleukin-4 gene 70 bp variable number tandem repeat polymorphism is associated with increased risk for FM [72].

Chemokines are a similar class of inflammatory mediators that transduce signals through G-protein-coupled receptors. The *CCL11* (C-C motif chemokine ligand 11) gene on chromosome 17 encodes a chemokine that can dampen the body’s response to inflammatory triggers. In a GWAS involving the FM family, Zhang et al. found that a SNP at *CCL11* (rs1129844) is associated with increased susceptibility to the development of FM [99]. Further, about 36% of the SNP was transmitted from parents to children, both of whom developed FM, and individuals with the SNP were found to have compensatory increases in the expression of CCL11. This suggests an underlying immune connection to FM. The *RNF123* (Ring Finger Protein 123) gene on chromosome 3 encodes E3 ubiquitin-protein ligase [73], which plays a role in cell cycle progression, innate immunity, and metabolism. In a GWAS, a *RNF123* (ring finger protein 123) SNP (rs1491985) was associated with developing FM [21]. Lastly, lower levels of the anti-inflammatory agent alpha-one antitrypsin [100] were found in a survey of FM individuals in 10 countries [101]. Blanco et al. reported that a polymorphism in the *AAT* (Alpha-1-antitrypsin) gene (PI*X) is found at higher frequencies in FM cohorts compared to the general population [101].

BDNF (brain derived neurotrophic factor) is a protein that regulates neuronal excitability and plasticity at multiple levels of the nervous system and has been shown in mouse models to play a key role in inflammatory pain and the development of chronic pain [76]. When released from the dorsal root ganglia, it acts on TrkB (tropomyosin receptor kinase B) receptors on primary afferent nerve endings and post-synaptic tracts in the spinal cord to amplify and potentiate ascending sensory signals. At the level of the periaqueductal grey matter, the BDNF-TRKB system is involved in the pathophysiologic mechanisms underlying several anxiety and depressive disorders and is the target of several antidepressant drugs [75]. Altered BDNF levels in the blood and cerebrospinal fluid are thought to play a role in the pathophysiology of FM, and these altered levels are largely genetically determined. A *BDNF* polymorphism (rs12273539) is associated with susceptibility to FM and symptoms of FM [74]. The rs7124442 and rs2049046 *BDNF* polymorphisms are associated with a higher body mass index and anxiety symptoms in FM individuals [78]. Similarly, the *BDNF* Val66Val SNP is associated with elevated plasma levels of high-sensitivity C-reactive protein and a higher body mass index [79]. Lastly, a *BDNF* polymorphism (rs6265) likely modulates pain signals at the level of the periaqueductal grey matter and is associated with pain catastrophizing in FM individuals [30].

Vitamin D plays a role in regulating nociceptive and inflammatory pain [80]. Vitamin D expression and regulation differs in patients with FM. The Apal and Fokl polymorphisms of the Vitamin D receptor (VDR) gene are associated with the development of FM [102]. Specifically, women with the C allele for Apal polymorphism are 3.33 times more likely to have FM, and women with the T allele for Fokl polymorphism are 10.9 times more likely to have FM. Further, Balkarli et al. showed that haplotypes of VDR gene polymorphisms are risky (ATC and TTT) and protective (TTC) for FM [17]. In addition, cannabinoid receptors in the body are implicated in the mood symptoms of FM. A FM family carrying the rs6454674 SNP in the cannabinoid receptor gene (CNR1) is associated with the development of depression [60].

### 4.5. Polymorphisms in Mitochondrial DNA and Vascular Genes and FM

In terms of metabolic changes and FM (Table 3), Tilburg et al. reported that the C allele at the SNP m.2352T>C (rs28358579) in the mitochondrial DNA is associated with increased risk for FM with an odd ratio of 4.6 [82]. In their cellular study, they demonstrated that the SNP decreased mitochondrial membrane potential under conditions that required oxidative phosphorylation. Polymorphism in the gene encoding methylenetetrahydrofolate reductase (MTHFR) may also play a role in the development of FM. Deveci et al. reported that the *MTHFR* C677T genotype may be associated with increased FM risk and symptoms of stiffness and dry eyes [83].

As for vascular changes and FM, two mediators of vascular tone, endothelin-1 (EDN-1) and angiotensin-converting enzyme (ACE), are associated with FM. EDN-1 is a potent vasoconstrictor, and a polymorphism of *EDN1* (rs1800541) is associated with higher plasma levels of EDN1 in FM patients compared to control and may increase risk of developing FM [68]. Individuals with ACE I/D polymorphisms have been more susceptible to the development of FM [103].

## 5. Epigenetic Contributions to the Development of FM

### 5.1. Associations of DNA Methylation Changes and FM

DNA methylation describes a phenomenon where a methyl group is added to the C-5 position of the cytosine ring of DNA by methyltransferases, resulting in gene repression [77]. There are multiple reports that DNA methylation is associated with development of FM (Table 4). Achenbach et al. reported that DNA methylation at the binding sites for the transcription factors Sp1 and c/EBPalpha is higher in FM women with widespread pain syndrome. [81]. Gerra et al. reported that individuals with FM have hypermethylation in the metabotropic glutamate receptor 2 (GRM2) gene, resulting in altered pain processing [86]. Polli et al. reported that FM individuals have a higher serum level of BDNF than controls, which correlated with hypomethylation of *BDNF* [90]. Menzies et al. reported six differentially methylated genes in FM individuals compared to healthy controls. These genes are *BDNF, NAT15* (N-acetyltransferase 15), *HDAC4* (Histone deacetylase 4), *PRKCA* (Protein kinase C alpha), *RTN1* (Reticulon 1a), and *PRKG1* (Protein kinase cGMP-dependent 1a), which are important for neuron differentiation/nervous system development, skeletal/organ system development, and chromatin compaction [91]. Lee et al. studied the epigenetic association between adverse childhood experiences such as rape, physical abuse, and emotional abuse and the development of FM. RNA sequences and DNA methylation analysis showed that *DAP3* (death associated protein 3) and miR2100 were hypermethylated in FM individuals with adverse childhood experiences compared to FM individuals without adverse childhood experiences. DAP3 plays a key role in cell respiration and apoptosis [92]. Andrade et al. identified 1610 differentially methylated sites by comparing the DNA methylation profiles of FM individuals and healthy controls. 69% of these sites were found to be hypomethylated, and these genes are involved in signal transduction and calcium signaling in the MAPK (mitogen-activated protein kinase 1) signaling pathway, regulation of the actin cytoskeleton, endocytosis, and neuroactive ligand-receptor pathways. In a 281-twin individual epigenome-wide analysis of DNA methylation [104], Burri et al. reported that the CpGs loci with significant p-values were malate dehydrogenase 2, tetranectin, and heat shock protein beta-6 [105].

### 5.2. Micro-RNA and FM

Micro-RNA inhibits gene expression by direct mRNA cleavage or translational repression [106]. Multiple micro-RNAs have been found to be associated with FM development or FM related symptoms (Table 5). Hussein et al. reported that FM individuals had significantly increased levels of micro-RNA-320-a compared to healthy controls. Within the FM population, micro-RNA-320-a was found to be higher in individuals with insomnia, chronic fatigue syndrome, persistent depressive disorder, and primary headache disorder [107]. Similarly, Akaslan et al. reported that the plasma levels of micro-RNA-320a, micro-RNA-320b, and micro-RNA-142-3p are positively correlated with symptom severity score in general health, functional status, and mental symptoms in the FM impact questionnaire in women with FM [108]. Conversely, Bjersing et al. reported higher levels of micro-RNA-320a in FM individuals than controls, and these levels were inversely correlated with pain [109]. Iannuccelli et al. reported a significant negative correlation between micro-RNA-320-b and depression scores in women with FM compared to healthy controls [110]. Erbacher et al. reported that micro-RNA-182-5p targets bone morphogenic protein receptor 2 (BMPR-2) and interleukin 6 (IL-6) in FM patients. This is evidenced by increased levels of micro-RNA-182-5p and decreased BMPR-2 and IL-6 [111]. Sommer et al. reported that the micro-RNA-103a/107 family is correlated with pro-inflammation and pain severity [112]. Braun et al. reported an association between upregulated gene expression of microR103a and micro-RNA-107 and adaptive coping in FM individuals [113]. Masotti et al. analyzed microRNA in the blood and saliva of FM individuals and healthy controls. They discovered that six micro-RNAs (micro-RNA-23a-3p, micro-RNA-1, micro-RNA-133a, micro-RNA-346, micro-RNA-139-5p, and micro-RNA-320b) were downregulated in FM individuals. These micro-RNAs are involved in brain function/development and muscular functions [114]. Leinders et al. identified that higher levels of micro-RNA-let-07d correlated with reduced small nerve fiber density in FM individuals [115]. It is interesting that insulin-like growth factor-1, the downstream target of miR-let-07d, is aberrantly expressed in the skin of FM individuals with small nerve fiber impairment [116], supporting previous studies that the insulin-like growth factor-1 receptor plays an important role in peripheral nerve regeneration.

Through the use of genome-wide expression profiling of micro-RNAs, Cerda-Olnedoi et al. found five downregulated micro-RNAs (hsa-micro-RNA223-3p, hsa-micro-R451a, hsa-micro-RNA338-3p, hsa-micro-RNA143-3p, hsa-micro-RNA145-5p, and hsa-micro-RNA-21-5p) in FM individuals compared to healthy controls [117]. Their targets have not been elucidated. Bjersing et al. profiled micro-RNA in FM individuals using cerebral spinal fluid. They discovered that nine micro-RNAs’ (micro-RNA-21-5p, micro-RNA-145-5p, micro-RNA-29a-3p, micro-RNA-99b-5p, micro-RNA-125b-5p, micro-RNA-23a-3p, 23b-3p, micro-RNA-195-5p, micro-RNA-223-3p) expression levels were significantly lower than in healthy controls. Of the nine, micro-RNA-145 is associated with individuals with higher pain and fatigue levels [118].

### 5.3. Upregulation/Over-Expression of Genes and FM

Overexpression of inflammatory genes has been proposed as a crucial factor in the development of FM (Table 6). Han et al. evaluated serum protein expression profiles of FM individuals and healthy controls [119]. They identified twenty-two differentially expressed proteins between the groups. Using a decision tree model, they were able to differentiate FM individuals from healthy controls based on the expression levels of METTL18 (Methyltransferase like 18), IGL3-25 (Immunoglobulin light 3-25), and Il1RAP (Interleukin 1 receptor accessory protein) with an accuracy of up to 97 percent. These proteins were associated with the upregulation of inflammatory pathways. Dolcino et al. investigated the gene expression profile in peripheral blood mononuclear cells from FM individuals and healthy controls [120]. They reported that genes involved in the immunologic pathway connected to interleukin-17 and type I interferon were overexpressed. This was confirmed by qPRC and antibody testing. Their findings suggest an autoimmune origin for FM. Similarly, Jones et al. analyzed RNA expression in FM patients and healthy controls and reported notable up-regulation of genes in inflammatory pathways and pain processing, such as glutamine/glutamate signaling and axonal development [121].

Fatima et al. studied the role of inflammation in FM by measuring the inflammatory cytokines in FM individuals and healthy controls [122]. They reported that the levels of TNF-alpha (tumor necrosis factor-alpha), IL-6, and interleukin 1 beta were significantly higher in FM individuals. These levels were positively correlated with clinical symptoms seen in FM individuals. In a similar study, Salemi et al. also detected higher expression of interleukin-1beta, IL-6, and TNF-alpha in the skin of FM individuals compared to healthy controls [123]. Pernambuco et al. evaluated the plasma levels of pro- and anti-inflammatory cytokines in FM individuals and healthy controls [124]. They found that interleukin-17A levels were significantly higher in FM individuals, and this positively correlated with levels of interleukin-2, interleukin-4, interleukin-10, TNF, and interferon-gamma. In contrast to the above, Surendran et al. did not find any significant differences in serum IL-6 and IL-8 in an Indian FM population compared to healthy controls [125], suggesting that there is a heterogeneous cytokine profile in different FM ethnic groups. Additionally, Erbacher et al. found that a miRNA targets IL-6 in FM individuals, resulting in decreased IL-6 production [111]. They postulated that the miRNA may have acted in a compensatory manner to confine an already pro-inflammatory state. If this is true, their findings support the hypothesis that inflammation increases FM risk.

In a proteomic analysis, Ramirez-Tejero et al. showed a protein expression pattern that favored biological pathways related to inflammation [126]. Specifically, the complement and coagulation cascades were activated in the FM individuals compared to healthy controls.

Trying to identify biomarkers of FM, Ilgun et al. evaluated the correlations between the neutrophil/lymphocyte ratio (NLR) and platelet/lymphocyte ratio (PLR) and the number of tender points on a physical exam [127,128]. They showed that PLR levels were significantly higher in FM individuals compared to healthy controls, and there was a positive correlation between PLR levels and the number of tender points. The NLR levels were similar in both groups, and there were no correlations. Stensson et al. investigated lipid mediators associated with anti-inflammatory effects in individuals with chronic widespread pain [129]. They found elevated levels of oleoylethanolamide and palmitoylethanolamide, which may serve as biomarkers for chronic musculoskeletal pain and are an indication of systemic inflammation. In terms of a pain marker in FM, Korucu et al. reported that FM individuals had higher levels of calcitonin gene-related peptide and expression of its receptors, the calcitonin receptor-like receptor and receptor component protein [130]. CGRP has been shown to be responsible for the development of peripheral and central sensitization [131].

Fineschi et al. compared the gene expression profile for peripheral B-cells and a panel of inflammatory serum proteins in FM individuals and controls [132]. RNA sequencing showed increased expression of interferon-regulated genes (S100A8 and S100A9, VCAM, CD163, SERPINA1, ANXA1) and inflammatory proteins (IL8, AXIN1, SIRT2, and STAMBP). These findings suggest an interferon signature in B-cells and an associated increase in a subset of inflammatory proteins in FM.

Neuroinflammation may play a role in the development of FM. Lo et al. used Diffusion Kurtosis Imaging (DKI) to visualize the microstructural alterations associated with neuroinflammation in specific brain regions of FM individuals compared to controls [133]. They reported that FM individuals had significantly higher levels of Abeta1-42 than controls. The DKI values were significantly lower in the bilateral dorsolateral prefrontal cortex and orbitofrontal cortex. In an evaluation of cerebral spinal fluid, Khoonsari et al. performed quantitative analysis of cerebral spinal fluid proteins obtained from FM individuals, rheumatoid arthritis individuals, or healthy controls [134]. They reported that proteins related to synaptic transmission, inflammatory responses, neuropeptide signaling, and hormonal activity were higher in FM and rheumatoid arthritis patients than controls.

Kaufmann et al. investigated whether stress hormones and the endocannabinoid anandamide affect neutrophil function in FM individuals [135]. Comparing FM individuals to controls, they found that the plasma levels of catecholamines and anandamide were significantly higher in FM individuals. This also correlated with increased hydrogen peroxide release, which is a marker of neutrophil function.

Salemi et al. investigated the opioid receptors expression in FM individuals compared to controls [136]. mRNA expression showed upregulation of the delta-opioid and kappa-opioid receptor genes and downregulation of the mu-opioid receptor gene. This was confirmed by qPRC and skin biopsy specimens. Bennett et al. assessed the relationship between the Nielsen test, digital photoplethysmography, and alpha 2-adrenergic receptors in FM individuals with cold intolerance and Raynaud’s syndrome [137]. They found that upregulation of alpha 2-adrenergic receptors in a subgroup of FM individuals positively correlated with symptoms of vasospasm. They postulated that increased expression of alpha 2-adrenergic receptors may be the cause of an exaggerated reaction to cold.

In a differential gene expression analysis between FM individuals and controls, Lukkahatai et al. reported that 57 genes can be used to differentiate FM individuals from non-FM individuals with a validation accuracy of 85.11% [138]. These genes are implicated in hepatic stellate cell activation, oxidative phosphorylation, and airway pathology related to chronic obstructive pulmonary disease.

Zhang et al. compared eotaxin and MCP-1 levels in FM individuals and controls [139]. They reported that FM individuals had higher levels of eotaxin compared to controls. In a migration study, eosinophils and monocytes migrated further when exposed to diluted plasma obtained from FM individuals versus controls. Similarly, Furer et al. reported that FM individuals had higher levels of Eotaxin-2 than controls [140].

Heat shock proteins are molecular chaperones that contribute to cellular homeostasis by keeping proteins in a functional conformation state [141]. The synthesis of heat shock protein is known to increase when the cells are exposed to stress [142]. In an ELISA (enzyme-link immunosorbent assay) comparing Caucasian women with FM to healthy controls, the FM individuals had significantly higher levels of HSP99AA1 (heat shock protein). Higher plasma HSP90AA1 were associated with higher number of pain sites, physical and emotional, fatigue, and reduced motivation [143]. In agreement with this, Majors et al. reported significant up-regulation of HSP90AA1 in FM individuals compared to controls [144]. In another study comparing gene expression patterns of women with FM and healthy race- and age-matched controls, Lukkahatai et al. They reported that HSP90AA1 was significantly elevated in the FM women [145]. When comparing the levels of HSP99AA1 between Caucasian and non-Caucasian women with FM to those of controls, the Caucasian FM women had a lower expression of HSP99AA1 [146]. Similarly, Caucasian FM men had lower plasma levels of HSP99AA1 than matched controls.

Pernambuco et al. compared 6-sulphatoxymelatonin (6-SMT) levels in the urine of FM individuals and controls [147]. 6-SMT is the main metabolite of melatonin. It is used as a surrogate measurement for the blood melatonin concentration. Pernambuco et al. reported that 6-SMT levels in the FM individuals were significantly higher, but there was no correlation with FM symptom severity. Measuring plasma protein, Kocak et al. reported that serum levels of Cathespin S (CatS) and Cystatin C (CysC) were higher in FM patients compared to controls [148].

**Table 6 biomedicines-11-01119-t006:** Associations of upregulated genes/proteins and FM.

Study	Genes/Proteins	Main Results
[68]	SNP (rs1800541) *EDN1* gene	FM patients had higher plasma levels of EDN1, a potent vasoconstrictor, compared to controls
[119]	*METTL18*, *IGL3-25*, *IL1RAP*, *IGHV1OR21-1*	Increased *METTL18*, *IGL3-25*, *IL1RAP*, and *IGHV1OR21-1* expression can differentiate FM individuals from healthy controls.
[120]	*Th-17* related genes (14 total), *Type 1 IFN* related genes (15 total)	Genes involved in the immunologic pathway connected to interleukin-17 and type I interferon were overexpressed in FM individuals compared to controls.
[121]	421 genes exhibited differential expression in FM patient compared to healthy controls	The genes identified are involved in pain processing and axonal development.
[122]	TNF-α, interleukin-1β, interleukin-6	Significant positive correlation between TNF-α, interleukin-1β, and interleukin-6 overexpression and FM symptoms.
[123]	TNF-α, interleukin-1β, interleukin-6	TNF-α, interleukin-1β, and interleukin-6 were upregulated in the skin tissue of FM individuals.
[124]	Interleukin-17A, interleukin-2, interleukin-4, interleukin-10, TNF, interferon necrosis-gamma	IL-17A levels were significantly higher in FM individuals.
[125]	Interleukin-1Ra, interleukin-8, interleukin-6	FM individuals had lower serum Interleukin-1Ra levels and normal interleukin-8 and interleukin-6 levels.
[126]	33 genes were overexpressed	Interplay between inflammation, coagulation, and complement cascades contributes to an inflammatory state in FM individuals compared to controls.
[128]	NLR, PLR	Systemic inflammatory response marker PLP levels correlated with the number of tender points in FM individuals.
[129]	Oleoylethanolamide, palmitoylethanolamide	Plasma levels of oleoylethanolamide and palmitoylethanolamide were significantly higher in FM individuals than in controls. Both are potential indicators of systemic inflammatory state in chronic widespread pain individuals.
[130]	CGRP, CLR, RCP	CGRP, CLR, and RCP were elevated in FM versus controls.
[132]	S100A8, S100A9, VCAM, CD163, SERPINA1, ANXA1, interleukin-8, AXIN1, SIRT2, STAMBP	Overexpression of identified proteins are associated with an interferon signature in B cells and increased inflammation in FM individuals.
[133]	Abeta1-42	Abeta1-42 was significantly higher in FM individuals.
[134]	Neural cell adhesion molecule L1, complement C4-A, lysozyme C, receptor-type tyrosine-protein, phosphatase zeta, apolipoprotein D, alpha-1-antichymotrypsin granulins, calcium/calmodulin-dependent protein kinase type II subunit alpha, mast/stem cell growth factor receptor, prolow-density lipoprotein receptor-related protein 1	Identified proteins were higher in FM and rheumatoid arthritis patients than controls. These proteins are related to synaptic transmission, inflammatory responses, neuropeptide signaling, and hormonal activity.
[135]	Catecholamines, anandamide	Plasma levels of catecholamines and anandamide are higher in FM individuals.
[136]	Delta-opioid receptor, Kappa-opioid receptor, mu-opioid receptor	Upregulation of delta and kappa receptor and downregulation of mu receptors in FM individuals compared to controls.
[137]	Alpha 2-adrenergic receptors	Upregulated alpha 2-adrenergic receptors in FM individuals positively correlate with receptor number and vasospastic symptoms.
[138]	57 genes linked to hepatic stellate cell activation, oxidative phosphorylation, and airway pathology related to COPD	Expression of these genes can differentiate FM individuals from healthy controls with high validation accuracy.
[139]	Eotaxin, MCP-1	Elevated levels of Eotaxin and MCP-1 in FM individuals versus controls.
[140]	Eotaxin-2	Higher levels of Eotaxin-2 in FM versus controls, but no correlation between marker levels and FM disease severity.
[143]	HSP99AA1	High levels are expressed in FM individuals; higher plasma levels are associated with an increased number of pain sites, fatigue, and decreased motivation.
[144]	HSP99AA1	HSP99AA1 was significantly upregulated in FM individuals.
[145]	12 genes identified, including CENPK, HSP99AA1	CENPK and HSP99AA1 were significantly elevated in FM women.
[147]	6-SMT	6-SMT was significantly elevated in FM, but there was no correlation with disease severity.
[148]	CatS, CysC	Serum levels of CatS and CysC were higher in FM individuals than controls.

*METTL18*: methyltransferase like 18; *IGL3-25*: immunoglobulin light 3-25; *Il1RAP*: interleukin 1 receptor accessory protein; TNF-α: tumor necrosis factor alpha; NLR: neutrophil/lymphocyte ratio; PLR: platelet/lymphocyte ratio; CGRP: calcitonin gene-related peptide; CLR: calcitonin receptor-like receptor; RCP: receptor component protein; S100A9: S100 calcium binding protein A9; VCAM: sascular cell adhesion molecule 1; SERPINA1: Serpin family A member 1; ANXA1: annexin A1; AXIN1: axin 1; SIRT2: sirtuin 2; STAMBP: STAM binding protein; HSP99AA1: heat shock protein 99A1; 6-SMT: 6-sulphatoxymelatonin; S100A8: S100 calcium binding protein A8; *EDN1*: endothelin 1; CENPK: centromere protein K; CatS: cathespin S; CysC: cystatin C.

### 5.4. Associations of Down-Regulated Genes/Proteins and FM

Like up-regulation of genes, down-regulation of genes may play a role in the development and pathophysiology of FM (Table 7). ɤ-aminobutyric acid (GABA) is an inhibitory neurotransmitter of the central nervous system. It has been proposed that decreased GABA levels may result in chronic pain conditions. Foerster et al. demonstrated that GABA levels were reduced in the right anterior insula of FM individuals using 3T proton magnetic resonance spectroscopy during pressure-pain testing [149]. Reduced GABA levels positively correlated with lower pressure-pain thresholds in the FM individuals.

An inflammatory state due to reduced anti-inflammatory cytokines production can contribute to the development of FM. Keskin et al. examined the anti-inflammatory cytokine profiles in FM individuals and healthy controls. They reported significantly lower levels of serum IL-13 in FM patients [150]. Uceyler et al. conducted a similar study and reported decreased expression levels of IL-4 and IL-10 in FM individuals compared to healthy controls [151].

Cordero et al. investigated the relationship between inflammation, oxidative stress, and mitochondrial dysfunction [152]. They showed that TNF-alpha was elevated in FM individuals compared to healthy controls. FM individuals had higher levels of mitochondrial reactive oxygen species and reduced co-enzyme Q10. They postulated that inflammation in FM individuals could be dependent on mitochondrial dysfunction. Similarly, evaluating the correlations of oxidative and antioxidative parameters and FM symptoms severity, Fatima et al. reported reduced enzymatic activity of catalase, glutathione peroxidase, and glutathione reductase in FM individuals compared to controls [153]. Individuals with increased oxidative stress had more severe FM-related symptoms.

Interestingly, even though HSP90AA1 has been found to be elevated in FM individuals [142,143,144], Lukkahatai et al. reported that the levels of HSP99AA1 between Caucasian and non-Caucasian women with FM were lower than those of controls, and the Caucasian FM women had a lower expression of HSP99AA1 [146]. Similarly, Caucasian FM men had lower plasma levels of HSP99AA1 than matched controls.

Other proteins with reduced expressions in FM individuals were reported. Tas et al. showed via ELISA that the early growth response (EGR) protein family was lowered in FM individuals compared to healthy controls [154]. Uceyler et al. reported reduced opioid receptor binding to F-18-fluoro-ethyl-diprenorphine in the mid cingulate cortex using a PET scanner on FM individuals compared to controls [155]. Not surprising, the FM individuals also have decreased anti-inflammatory cytokine IL-4 levels compared to controls. Verma et al. observed that there were significantly fewer natural killer cells in FM individuals compared to controls via flow cytometry of peripheral blood mononuclear cells [156]. Interestingly, the natural killer cells obtained from FM individuals more responsive when co-cultured with human leukocyte antigen null target cells compared to healthy controls. Olama et al. found that FM individuals with lower serum leptin levels had decreased FM symptom severity [81].

**Table 7 biomedicines-11-01119-t007:** Associations of downregulated genes/proteins and FM.

Study	Genes/Proteins	Main Results
[133]	TNF-alpha	Elevated TNF-alpha levels in FM individuals are associated with higher levels of mitochondrial reactive oxygen species and reduced coenzyme Q10 activity.
[149]	GABA	FM individuals had reduced GABA levels in the right anterior insula compared to controls. This is positively correlated with lower pressure pain thresholds.
[150]	Interleukin-13	FM individuals had lower interleukin-13 levels than controls.
[151]	Interleukin-4, Interleukin-10	FM individuals had decreased expressions of interleukin -4 and interleukin-10 compared to controls.
[154]	EGR1	FM individuals had lower serum EGR1 compared to controls.
[155]	HSP99AA1	Gender and race may alter levels of HSP99AA1in FM individuals; HSP99AA1 levels lower in non-Caucasian FM individuals; and HSP99AA1 levels lower in Caucasian men with FM compared to controls
[155]	Interleukin -4, Interleukin -10, Opioid receptor	FM individuals had reduced opioid receptor binding to F-18-fluoro-ethyl-diprenorphine in the mid cingulate cortex compared to controls. FM individuals had low interleukin-4, interleukin-10 gene expression levels.
[156]	Natural killer cells	FM individuals had less natural killer cells than controls, but it was more responsive when exposed to human leukocyte antigen null target cells.
[122]	Catalase, glutathione peroxidase, glutathione reductase	Identified enzymes were significantly lower in FM; expression levels correlated with higher oxidative stress parameters compared to controls; and correlated with severity of FM related symptoms.

GABA: ɤ-aminobutyric acid; EGR1: Early Growth Response 1.

## 6. Conclusions

The genetic and epigenetic etiologies of FM remain elusive. There is no direct genetic variant or epigenetic modification that results in the development of FM. Genetic studies focusing on polymorphisms in genes involved in the catecholaminergic and serotonergic pathways reveal mixed outcomes. Depending on the population studied, the genes involved may either be associated with an increased risk of developing FM or no risk. Similarly, it is unclear whether these gene polymorphisms are associated with the severity of FM symptoms.

Meanwhile, genetic studies of polymorphisms in genes involved in pain processing, inflammation, and oxidative stress suggest an underlying connection to the severity of FM symptoms. Further studies are needed to evaluate whether these relationships exist in diverse populations.

DNA methylation studies comparing FM individuals to healthy controls or in twins have suggested that certain methylated genes may be associated with the development of FM. However, there are no molecular explanations of how this leads to FM susceptibility. Likewise, many microRNAs have been found in FM individuals compared to controls, but their targets and effects are still unknown. Given the studies of up/down regulated genes and its association with FM development and symptoms severity, future studies may need to examine if there are direct connections between the epigenetically modified genes and these up/down regulated genes.

Regarding the differential gene expressions found in FM individuals, a possible molecular and cellular explanation may be related to the factors that influence epigenetic modifications, such as diet, medical co-morbidities, psychological stressors, physical activity, environmental pollutants, and sleep [157]. Depending on the temporal relationships between FM and these factors, the gene expression profile may vary. Additionally, the specimen obtained for the studies may not reflect FM physiology. Many studies that have investigated gene expression changes use peripheral blood samples, and information obtained from these samples may not reflect the changes in the CNS and PNS. FM is a central sensitization phenomenon. Moreover, histone modification, which is a major epigenetic modulator, in FM has not been reported. Future studies need to expand the current knowledge of microRNAs and DNA methylation and investigate the relationship between histone modification and FM.

In summary, there is no gene that results in the development of FM. It may be a combination of genetics, epigenetics, and environmental changes that lead to the development of FM. The combinations may change depending on the genetic make-up of the individual, medical co-morbidities, psychosocial stressors, and cultural and religious influences.

### Future Directions

This review focuses on the genetic and epigenetic contributions to the development of FM. There are other areas of research that may be considered in the future. The mechanism underlying fatigue in FM is still poorly understood. There is discord in the current literature as to whether fatigue is a centrally mediated process or the result of differences in the percentage of type 1 muscle fibers [3]. It is worth noting that fatigue in FM occurs more frequently in women, and gender differences may be a good area for future research. Indeed, sexually dimorphic dorsal root ganglia have been discovered between genders, and research is ongoing as to whether differences in the nervous system explain the increased prevalence of FM in women [158]. The role of increased glutamatergic tone in centrally mediated pain and drugs that modulate NMDA receptor activation are another area of research that should be further explored. Furthermore, genome wide association studies thus far have identified several FM-associated genetic variants the rs11127292 SNP at the MYT1L locus involved in neuronal differentiation and a copy number variation in the NRXN3 locus involved in cell adhesion. Despite these variants not being statistically significant, further studies may consider using FM individuals from different ethnic backgrounds [159].

## Figures and Tables

**Figure 1 biomedicines-11-01119-f001:**
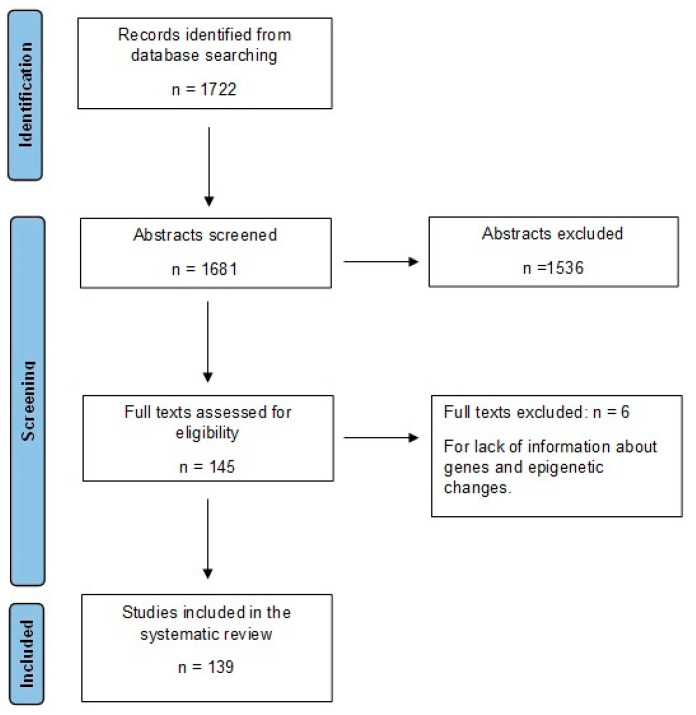
PRISMA (Preferred Reporting Items for Systematic Reviews and Meta-Analyses) flow diagram of the literature search and selection process for fibromyalgia, genetics, and epigenetics.

**Figure 2 biomedicines-11-01119-f002:**
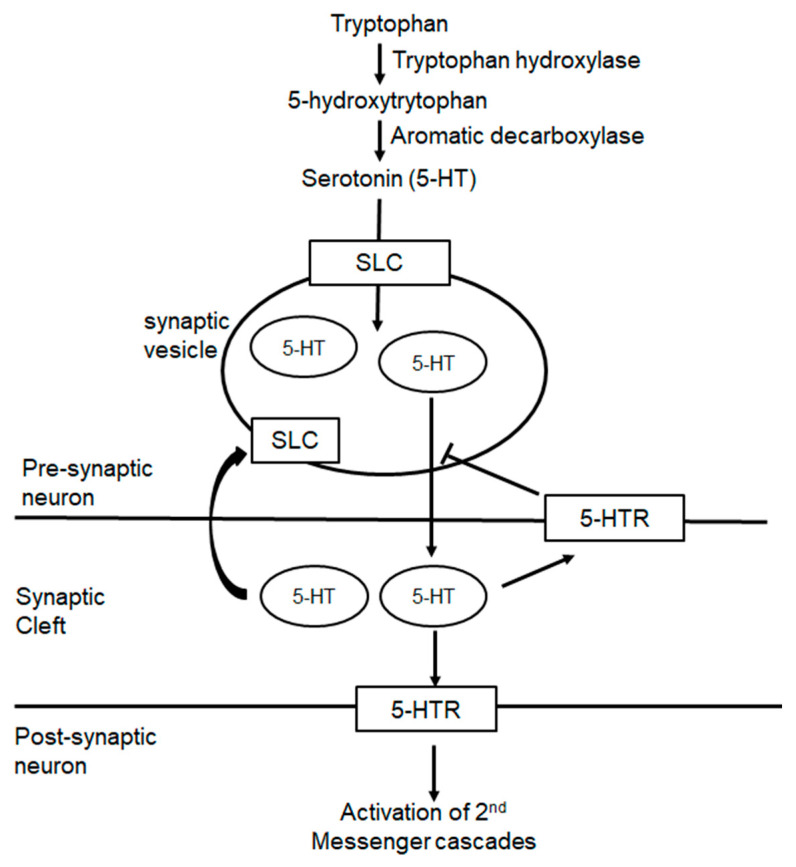
5-HT biosynthesis and physiology. 5-HT: serotonin; SLC: solute carrier; 5-HTR; serotonin receptor.

**Table 1 biomedicines-11-01119-t001:** Associations of COMT Polymorphisms and FM Development and Symptoms.

Study	Genes	Effects in FM
[12,32,33,34,35,36]	*COMT* Val-158-Met (rs4680)	Increased risk of developing FM
[12,36,42]	rs4818 of the *COMT* gene	No increased risk of FM development in diverse, Mexican, or Spanish populations
[32,36,38]	rs4818 of the *COMT* gene	Increased risk of FM development in Korean, Brazilian, and Spanish populations
[36]	rs6269 of the *COMT* gene	Increased risk of FM development in the Spanish population
[37,38,39,40]	*COMT* Val-158-Met (rs4680)	No increased risk of developing FM
[38]	rs4633 of the *COMT* gene	Increased risk of FM development in the Korean population
[42]	*COMT* Val-158-Met (rs4680)	Increased severity of symptoms
[43]	rs2097903 of the *COMT* gene	Increased risk of FM development

*COMT*: Catechol-O-Methyltransferase; FM: fibromyalgia.

**Table 2 biomedicines-11-01119-t002:** Associations of Genes in the Serotonin Pathway and FM Development and Symptoms.

Study	Genes (Polymorphisms)	Effects In FM
[41]	*5-HTR2A*, CT polymorphism genotype	Lower pain threshold
[50,51]	*SLC6A4* “short allele”	Increased risk of FM
[53]	*5-HT1a*, CC/G polymorphism alone and with 5-HTT-low	Increased depressive symptoms
	*5HT1a*, CC polymorphism with 5-HTT-high	Fewest depressive symptoms, highest response to an SSRI
[54]	*5-HTR3a*, SNP rs1062613 CC homozygote	Increased risk of FM
[55]	*5-HTR3a*, *5-HTR3b*	Unsure currently

*5-HTR2A*: 5-hydroxytryptamine receptor 2A; *SCL6A4*: solute carrier family 6 member 4; *5-HT1a*: serotonin 1A receptor; *5-HTR3a*: 5-hydroxytryptamine receptor 3A; *5-HTR3b*: 5-hydroxytryptamine receptor 3B; SSRI: Selective serotonin reuptake inhibitor, CT: Cytosine-Thymine.

**Table 3 biomedicines-11-01119-t003:** Associations between Polymorphisms in Pain Pathway Processing Genes, Inflammatory Genes, Mitochondrial DNA, and Vascular Genes and the Development of FM.

Study	Genes	Effects in FM
[21]	*ATP2C1*	Regulates calcium homeostasis through the hSPCA1 magnesium-dependent calcium channel and affects nerve function.
[60]	*HAP1*	Regulates vesicular trafficking and lowers the nociceptive flexion reflex threshold.
[61]	*SCN9A*	Encodes the Nav1.7 sodium channel; the rs4796604 SNP associated with reduced motivation, reduced activity, and higher FM impact questionnaire scores.
[62]	*OPRM1*	Modulates cerebral pain processing.
[63,64]	*TRPV3*	Plays a role in the sensation of noxious stimuli; the rs395357 SNP increases the symptom severity of fatigue and mental health in FM.
[65]	*TRPM6*	Regulates calcium and magnesium homeostasis in neurons.
[66,67]	*SNAP25*	Regulates neurotransmitter release via vesicle docking and fusion; TC genotype is associated with higher pain and depression scores
[68]	*GCH1*	Involved in dopamine and serotonin synthesis. rs3783641, rs84, rs752688, and rs4411417 SNPs are associated with lower pain sensitivity in FM.
[69]	*TAAR1*	Encodes G-protein-coupled receptor and plays a role in reward and cognitive function. Increased levels decrease reward seeking behavior and increase attention and focus.
[70]	*RGS4*	Encodes a GTPase that interacts with PAR4 GPCR and negatively regulates G-protein signaling.
[71]	*GRIA4*	Encodes a glutamate receptor that mediates excitatory neurotransmission.
[72]	Beta2-Adrenergic Receptor gene	The Gly16Arg SNP increases the risk of FM and may cause sleep dysfunction in FM.
[73]	alpha(1A)-Adrenergic Receptor	rs1383914 and rs1048101 SNPs are associated with higher FM impact questionnaire scores; rs574584 is associated with a higher FM impact questionnaire score, increased stiffness, and increased fatigue.
[74]	*CCL11*	Associated with increased susceptibility to FM due to higher levels of plasma chemokines and an increased inflammatory response.
[75]	*VNTR*	Encodes interleukin 4. A 70 bp polymorphism at this locus is associated with a higher risk for FM.
[76]	*C11orf40*	Associated with higher levels of inflammatory cytokines.
[76]	*ZNF77*	Associated with higher levels of inflammatory cytokines.
[77]	*MEFV*	Encodes a protein called pyrin, which suppresses inflammation; missense mutations of this gene correlated with higher levels of plasma IL-1beta.
[78]	*RNF123*	Encodes E3 ubiquitin-protein-ligase, which plays a role in cell cycle progression, innate immunity, and the metabolism of proteins. rs1491985 SNP is associated with an increased risk of developing FM.
[79,80,81]	*AAT*	Encodes alpha-1-antitrypsin; the PI*Z polymorphism has increased prevalence in FM patients. ¼ to 1/36 of FM patients found to have AAT deficiency.
[32,82,83,84]	*BDNF*	rs12273539 SNP is associated with susceptibility and symptoms of FM; rs7124442 and rs2049046 SNPs are associated with body mass index and anxiety symptoms of FM. rs6265 polymorphism is associated with pain catastrophizing in FM. Val66Val SNP is associated with elevated plasma CRP and body mass index.
[67,85]	*VDR*	Encodes the vitamin D receptor; Apal polymorphism and Fokl polymorphism increase the risk of developing FM in women.
[86]	*CNR1*	Encodes the cannabis receptor; the rs6454674 SNP increases the risk for depression in FM.
[87]	*m.2352C*	Mitochondrial DNA gene, which increases the risk of developing FM.
[88,89,90,91]	*MTHFR*	Encodes a key enzyme in folate metabolism and increases the risk of developing FM.
[92]	*EDN-1*	Encodes endothelin-1, a potent vasoconstrictor; the rs1800541 SNP is associated with higher levels of endothelin-1 and susceptibility to FM.
[93]	*ACE I/D*	Encodes angiotensin-converting enzyme; the ACE I/D polymorphism increases susceptibility to FM.

*TRPM6*; transient receptor potential cation channel subfamily M member 6; *ATP2C1:* ATPase secretory pathway Ca^2+^ transporting 1; *BDNF*: brain derived neurotrophic factor; *HAP1*: Huntingtin associated protein 1; *TRPV3*: transient receptor potential vanilloid 3; *SNAP25*: synatposome associated protein 25; *VDR*: vitamin-D receptor; *SCN9A*: sodium voltage gated channel alpha subunit 9; *OPRM1*; opioid receptor mu 1; *TAAR1*: trace amine associated receptor 1; *RGS4*: regulator of G protein signaling 4; *GRIA4*: glutamate ionotropic receptor AMPA type subunit 4; C11orf40: chromosome 11 putative open reading frame 40; *ZNF77*: zinc finger protein 77; *VNTR*: variable nucleotide tandem repeat; *CCL11*: C-C motif chemokine ligand 11; *RNF123*: ring finger protein 123; *AAT*: alpha-1-antitrypsin; *CNR1*: cannabinoid receptive 1; *MTHFR*: methylenetetrahydrofolate reductase; *ACE I/D*: angiotensin-converting enzyme insertion/deletion;; *GCH1*: guanosine triphosphate cyclohydrolase 1; *MEFV*: mediterranean fever in a immunity regulator, pyrin; *EDN-1*: endothelin-1.

**Table 4 biomedicines-11-01119-t004:** Association of DNA methylation changes and FM.

Study	Genes	Main Results
[81]	*Sp1* *C/EBPalpha*	DNA methylation at Sp1 and C/EBPalpha correlated with widespread pain syndrome and decreased leptin expression and serum leptin levels in FM individuals.
[86]	*GRM2*	Hypermethylation of *GRM2* in FM individuals compared to healthy controls.
[90]	*BDNF*	Hypomethylation at exon 9 of the *BDNF* gene. BDNF levels are increased in FM individuals.
[91]	*BDNF*, *NAT15*, *HDAC4*, *PRKCA*, *RTN1*, *PRKG1*	Differentially methylated in these genes in women with FM compared to healthy controls. These genes are involved in neuron differentiation/nervous system development, skeletal/organ system development, and chromatin compaction.
[92]	*DAP3* miR2100	Hypermethylation at these sites in FM individuals with adverse childhood experiences compared to FM individuals without adverse childhood experiences.
[104]	69% of the differentially methylated genes are in the MAPK signaling pathway, regulation of the actin cytoskeleton, endocytosis, and neuroactive ligand receptor pathways.	FM individuals had 1610 differentially methylated positions compared to healthy controls.
[105]	*MDH2*, *CLEC3B*, *HSPB6*	In a 281-twin individual epigenome-wide analysis of DNA methylation, CpG loci with significant *p*-values were MDH2, tetranectin, and heat shock protein beta-6.

*GRM2*: metabotropic glutamate receptor 2; *BDNF*: brain derived neurotrophic factor; *NAT15*: N-acetyltransferase 15; *HDAC4*: histone deacetylase 4; *PRKCA*: protein kinase C alpha; *RTN1*: reticulon 1a; *PRKG1*: protein kinase cGMP-dependent 1a; *DAP3*: death associated protein 3; MAPK: mitogen-activated protein kinase 1; *MDH2*: malate dehydrogenase 2; *CLEC3B*: C-type lectin domain family 3 member B; *HSPB6*: heat shock protein B6.

**Table 5 biomedicines-11-01119-t005:** Associations of micro-RNA and FM.

Study	micro-RNAs	Targeted Genes	Main Results
[107]	micro-RNA-320a		FM individuals had significantly increased levels of micro-RNA-320-a compared to controls. Within the FM population, increased micro-RNA-320-a levels are associated with insomnia, chronic fatigue syndrome, persistent depressive disorder, and primary headache disorder.
[108]	micro-RNA-320a, micro-RNA-320b, micro-RNA-142-3p		The plasma levels of micro-RNA-320a, micro-RNA-320b, and micro-RNA-142-3p are positively correlated with the symptom severity score in general health, functional status, and mental symptoms in the FM impact questionnaire in women with FM.
[109]	micro-RNA-320a		FM individuals with increased levels of micro-RNA-320-a had decreased pain.
[110]	microRNA-320b		There is a significant negative correlation between micro-RNA-320-b and depression scores in women with FM compared to healthy controls.
[111]	micro-RNA-182-5p	Bone morphogenic protein receptor 2, interleukin 6	Micro-RNA-182-5p targets bone morphogenic protein receptor 2 and interleukin 6 in FM individuals.
[112]	micro-RNA103a/107		Micro-RNA-103a/107 is correlated with pro-inflammation and pain severity.
[113]	micro-RNA103a, micro-RNA107		Positive association between micro-RNA103a and micro-RNA-107 expressions and adaptive coping in FM individuals.
[114]	micro-RNA-23a-3p, micro-RNA-1, micro-RNA-133a, micro-RNA-346, micro-RNA-139-5p, and micro-RNA-320b		micro-RNA-23a-3p, micro-RNA-1, micro-RNA-133a, micro-RNA-346, micro-RNA-139-5p, and micro-RNA-320b were downregulated in FM individuals.
[115]	micro-RNA-let-7d	Insulin-like growth factor-1	Higher levels of micro-RNA-let-07d correlated with reduced small nerve fiber density in FM individuals. Insulin-like growth factor-1 is a downstream target of micro-RNA-let-07d that may lead to small nerve fiber impairment.
[117]	hsa-micro-RNA223-3p, hsa-micro-R451a, hsa-micro-RNA338-3p, hsa-micro-RNA143-3p, hsa-micro-RNA145-5p, and hsa-micro-RNA-21-5p		hsa-micro-RNA223-3p, hsa-micro-R451a, hsa-micro-RNA338-3p, hsa-micro-RNA143-3p, hsa-micro-RNA145-5p, and hsa-micro-RNA-21-5p are significantly downregulated in FM individuals.
[118]	micro-RNA-21-5p, micro-RNA-145-5p, micro-RNA-29a-3p, micro-RNA-99b-5p, micro-RNA-125b-5p, micro-RNA-23a-3p, 23b-3p, micro-RNA-195-5p, and micro-RNA-223-3p		Expressions of micro-RNA-21-5p, micro-RNA-145-5p, micro-RNA-29a-3p, micro-RNA-99b-5p, micro-RNA-125b-5p, micro-RNA-23a-3p, 23b-3p, micro-RNA-195-5p, and micro-RNA-223-3p are significantly lower in FM individuals compared with healthy controls. MiR-145 is associated with pain and fatigue.

## Data Availability

Not applicable.

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
