# Peer review of "A Comprehensive Review of the Genetic and Epigenetic Contributions to the Development of Fibromyalgia"

_biomedicines, 2023, doi:10.3390/biomedicines11041119_

Round 1
Reviewer 1 Report
I read with great interest of the article. The paper is very well-written and well-organization. Besides, this paper increases our knowledge and broad overview about the role of genetic and epigenetic in the development of fibromyalgia.
Minor concerns:
1. A single paragraph to describe the pathophysiology of fibromyalgia should be added in the text.
Reviewer 2 Report
The authors presented a really comprehensive review related to molecular and genetic aspects of FN development.
However, some issues should be corrected.
Comments
1. All the typos should be corrected. Line 530: The term “gender” should be used instead of “sex”.
2. Lines 50-51; 488-489; 523-524; 537-539: These sentences are not clear. They should be clarified.
3. Lines 494-495 and 273-274: All the repeats should be removed.
4. Lines 80-103: The review on 5-HT biosynthesis and physiology should be supplemented by schematic presentation(s).
5. Table 3 title should be modified as the term “other” is not clear.
6. Lines 318-319 versus line 361: The discrepancy between the levels of IL-6 expression in different studies should be discussed.
7. Lines 437-438 and line 376; line 452: The downregulated genes should be moved to the next section
8. Section 3.3 and 3.4: The authors should discuss the possible molecular and cellular mechanisms underlying differential gene expressions in FM.
Reviewer 3 Report
Thank you for the opportunity to review this interesting article. I consider that authors have performed a good and hard job. Congratulations.
I have only two minor concerns:
At first, I would consider to increase the length of introduction.
Secondly, although it was a narrative review, I considert that it is necessary a methods section where appears search strategy and the principal methodological steps to perform the review.
